# Overcoming the Limitations of Stem Cell-Derived Beta Cells

**DOI:** 10.3390/biom12060810

**Published:** 2022-06-09

**Authors:** Mariana V. Karimova, Inessa G. Gvazava, Ekaterina A. Vorotelyak

**Affiliations:** 1Koltzov Institute of Developmental Biology of Russian Academy of Sciences, 119334 Moscow, Russia; maryanna_karimova@mail.ru (M.V.K.); gvazava.inessa@yandex.ru (I.G.G.); 2Department of Biology, Lomonosov Moscow State University, 119991 Moscow, Russia

**Keywords:** insulin-producing cells, stem cell-derived beta cells, pancreas, diabetes, GSIS, microenvironment

## Abstract

Great advances in type 1 diabetes (T1D) and type 2 diabetes (T2D) treatment have been made to this day. However, modern diabetes therapy based on insulin injections and cadaveric islets transplantation has many disadvantages. That is why researchers are developing new methods to regenerate the pancreatic hormone-producing cells in vitro. The most promising approach is the generation of stem cell-derived beta cells that could provide an unlimited source of insulin-secreting cells. Recent studies provide methods to produce beta-like cell clusters that display glucose-stimulated insulin secretion—one of the key characteristics of the beta cell. However, in comparison with native beta cells, stem cell-derived beta cells do not undergo full functional maturation. In this paper we review the development and current state of various protocols, consider advantages, and propose ways to improve them. We examine molecular pathways, epigenetic modifications, intracellular components, and the microenvironment as a possible leverage to promote beta cell functional maturation. A possibility to create islet organoids from stem cell-derived components, as well as their encapsulation and further transplantation, is also examined. We try to combine modern research on beta cells and their crosstalk to create a holistic overview of developing insulin-secreting systems.

## 1. Introduction

The human pancreas comprises exocrine and endocrine tissue [1]. Endocrine cells are arranged into pancreatic islets, or islets of Langerhans, which include insulin-producing beta cells, glucagon-producing alpha cells, delta cells producing somatostatin, pancreatic polypeptide cells (PP cells) producing pancreatic polypeptide, and ghrelin-producing epsilon cells [2,3]. All types of cells work together in a complicated manner to maintain metabolic homeostasis. Further, their interactions play an important role in disease [4].

Diabetes mellitus affects millions of people all over the world. The cause of type 1 diabetes (T1D) is the loss of beta cells due to autoimmune response and inflammation [5]. Type 2 diabetes (T2D) is a metabolic disorder that develops because of beta cell dysfunction and is often characterized by insulin resistance [6]. Although major improvements in insulin delivery in type 1 diabetic patients have been achieved by using insulin pumps, glucose monitors, and an artificial pancreas, these methods can pose risks of hypo- and hyperglycemia, reduction of glycated hemoglobin, and other health complications in the case of comorbidities [7,8,9,10]. Allotransplantation of cadaveric islets is another approach to diabetes therapy. Transplantation followed by immunosuppression can result in long-term insulin independence [11,12]. However, this method is limited due to the lack of suitable donors. A variety of pharmacological treatment options are available for T2D, including glucagon-like peptide 1 (GLP1) receptor agonists and sodium–glucose cotransporter-2 (SGLT2) inhibitors, which help against hyperglycemia, despite having several drawbacks [13]. Such therapy for both T1D and T2D is still not available to some people and is an economic burden to healthcare systems [14,15].

Due to a pressing worldwide situation with diabetes, ideas to use direct differentiation from embryonic stem cells (ESCs) and pluripotent stem cells (PSCs) to produce beta cells have surfaced [16]. Stem cells are thought to be an ideal source of all cell types including pancreatic beta cells. However, beta cells are highly specialized; because of that, differentiation of mature insulin-secreting cells is extremely complex.

## 2. A Journey to Produce Functional Beta Cells

The methods of in vivo differentiation of beta cells mimicking the genetic and functional profiles of native cells have come a long way. One of the approaches is transdifferentiation, or cell reprogramming. Usually, the choice of the cell source for reprogramming is based on a common background and an availability of the cells. For pancreatic beta cells, those are the gallbladder [17], liver [18,19,20], exocrine [21,22] and other endocrine pancreatic cell types [23]. Reprogramming is achieved by overexpression of the genes involved in pancreas development and characterizing mature beta cells such as pancreatic and duodenal homeobox 1 (PDX1), neurogenin-3 (NGN3), member of the Maf family of transcription factors MAFA, paired box protein PAX6, paired box protein PAX4, and other transcriptional factors [24,25,26]. The aforementioned protocols have their advantages; however, they all pose the same question of how to achieve maximum transdifferentiation efficiency and minimize heterogeneity.

In this case, it seems more promising to use undifferentiated cells because it could be easier to direct them towards the beta cell identity [27]. Direct differentiation methods of acquiring mature beta cells from hESCs (human ESCs) and hPSCs (human PSCs) have been modified and improved. First, protocols for hESC-derived definitive endoderm have been developed [16]. This approach as the more recent ones involves supplementing the culture media with small molecules and growth factors such as activin A and fibroblast growth factor 10 (FGF10) that can potentiate hPSCs and hESCs towards functional beta cells [16,28]. The next step is differentiation of PDX1^+^ pancreatic progenitors from hESCs [29]. With further research, it has become possible to differentiate hPSCs into insulin-producing cells [30] and adapt the previous protocols using 3D clusters to create conditions more similar to the in vivo ones [31].

Methods to produce hESC- and hPSC-derived beta cells have been developing in parallel [32,33]. A protocol proposed by Rezania et al. [32] included seven stages of hESC differentiation. Each one imitated a stage in pancreas embryogenesis from the definitive endoderm to maturating beta cells. Cells at the last stage were characterized by the expression of MAFA, homeobox protein NKX6.1, insulin (INS), and the absence of glucagon (GCG), as well as glucose-stimulated insulin secretion (GSIS) [32]. Pagliuca et al. [33] implemented a similar method for hPSCs and generated functional insulin-secreting cells. Differentiated beta cells transplanted in streptozotocin (STZ)-induced diabetic mice [32] or NOD-Rag1^null^ IL2rg^null^ Ins2^Akita^ (NRG-Akita) mice [33] returned animals to normoglycemia. However, a close analysis of cells at the final stage of both protocols showed that although beta-like cells had comparable expression profiles and functional similarities to native cells, such as glucose-stimulated insulin secretion in vitro, there were major differences in insulin granule morphology, key transcription factor expression was lower, and GSIS dynamics were far from identical. Moreover, the cell population was heterogeneous as polyhormonal cells were present [32,33].

The protocols published over the past few years seem to face the same complications such as low differentiation efficiency, presence of polyhormonal cells, low expression rates or the lack of some beta cell markers, and, most importantly, functional immaturity of cells [32,33,34,35]. Some of these difficulties have been almost dealt with in more recent research [36]. However, it is still not possible to overcome all of them (Table 1).

It is worth mentioning that the search for progenitor cells in the adult pancreas is still ongoing, with a prospective that these cells could be used as another source for regenerative medicine. One example is a cell population that was identified in mice which expresses protein C receptor (Procr^+^ cells) [37]. It has been reported that adult murine islets contain Procr^+^ cells, which can differentiate into alpha, beta, delta, and PP cells. Pancreatic islet organoids derived from Procr^+^ cells can reverse hyperglycemia in diabetic mice.

However, the question of whether a population with the same properties and differentiation capacity exists in humans remains open. Interestingly, the expression of human endothelial protein C receptor (hEPCR) in donor murine islets improved the transplantation outcome [38]. Due to this fact, it might be perspective to further study protein C receptor functions. Despite the promising results, such findings should be addressed with caution because reports of progenitor populations in the adult pancreas have surfaced before, and the main drawback of such reports is that progenitor cells have an ability for clonogenic expansion only in vitro, while not showing the same properties in vivo [39]. The ways to generate beta cells are summarized in Table 1.

Direct differentiation can be used to generate insulin-secreting cells from human induced pluripotent stem cells (hiPSC) from patients with diabetes. T1D patient-derived hiPSCs serve as cell models for the anti-diabetic treatment effect, a search tool for new possible drugs, and for the assessment of different negative effects that could promote the onset of diabetes [40]. The fact that beta cells that were produced by Milman et al. [40] did not have any significant differences from stem cell-derived beta cell from non-diabetic donors can be a point of interest for the following research. Ones again, this can be the proof that diabetes is caused by not just one reason such as a mutation, but by several collective factors [41].

The progress that has been made in generating beta cells in vitro especially from human stem cells provides an opportunity for their application in regenerative medicine, studying beta cell identity, maturation, functions, as well as being tools for investigating disease and models for possible drug screening.

## 3. Addressing the Difficulties

As mentioned above, a major difficulty in producing beta cells is the fact that they lack something, yet to be identified, that native mature beta cells have. This might be the most perplexing question, probably without one simple answer. The issues related to low efficiency and heterogeneity have almost been dealt with [36]. The already proposed and possible advancements are discussed in the next paragraphs (Figure 1).

### 3.1. Efficiency of Differentiation and Polyhormonal Cells

The generated populations of cells often appear to be polyhormonal. Polyhormonal INS^+^/GCG^+^ cells are present during differentiation in vivo and in vitro [42]. In human fetal pancreas development, polyhormonal cells are noticeable as early as at week nine. The switch from the polyhormonal to monohormonal state is associated with lesser proliferative capacity and maturation [43]. Polyhormonal cells do not give rise to functional insulin-producing cells and appear to be immature alpha cells that only transiently express insulin [35,44]. INS^+^/GCG^+^ and INS^+^/SST^+^ cells (insulin^+^/somatostatin^+^ cells) differentiate from pancreatic progenitors (PPs) [33], so prolonging the PP stage improves the yield of INS^+^ cells, probably by providing more time for maturation [36].

Many protocols result in early endocrine commitment, which leads to low efficiency of functional beta cell generation [32,33,34]. To increase the number of INS^+^ cells, a closer investigation was performed considering the use of specific small molecules and growth factors during certain time frames for maximum effectiveness. For example, elimination of BMP inhibitors during pancreatic progenitor specification results in a higher yield of PDX1^+^/NKX6.1^+^ cells [34]. However, at a later stage of PDX1^+^/NKX6.1^+^ cells differentiation, BMP inhibitors promote NGN3 expression and as a result induce INS^+^/NKX6.1^+^ cells with a lower percentage of polyhormonal cells [34].

Liu et al. [36] proposed improvements to existing protocols such as creating an extended pancreatic progenitor 3D cluster culture and performing throughput analysis of combinations of applied chemicals. Using a cocktail of ten chemicals, with some of them never used before to produce functional beta cells from PPs, they generated NKX6.1^+^/INS^+^ hPSC-derived pancreatic beta cells with an efficiency up to 82%. These functional beta cells reversed hyperglycemia in diabetic mice in two weeks [36]. In vivo, beta cells gradually develop from progenitors and gain their distinctive features such as molecular and cell surface markers, as well as insulin secretion. Because these protocols imitate developmental processes in some way, prolonging some stages may allow cells to mature, and thus increase the efficiency of protocols.

Differentiation efficiency depends on the cell culture method as well. It was demonstrated that 3D cell clusters and the air–liquid interface technique yield the most prominent results [36]. An air–liquid interface promotes differentiation of insulin-producing cells from stem cells, and 3D cultures represent the in vivo conditions more closely [35].

### 3.2. Functional Maturation

The question of functional cell maturation is currently one of the most addressed, especially considering the beta cell. This topic is reviewed in a number of most recently published papers which focus on such aspects as physiological and genetic maturation strategies [45], signaling pathways that are required for this process [46], different maturation triggers, and metabolic networks [47]. In the mentioned reviews, it is once again emphasized that maturation does not depend on one signaling pathway, but that it involves overlapping intra-islet networks and crosstalk with microenvironment on different regulatory levels.

One prominent characteristic of mature beta cells is GSIS. Glucose is the most important stimulus in this process. Glucose influx increases metabolism in beta cells and results in higher levels of intracellular ATP. ATP-sensitive potassium channels (K_ATP_) close, which leads to membrane depolymerization and Ca^2+^ influx. The high levels of Ca^2+^ activate insulin granule fusion with cellular membranes and insulin release [48,49]. This is a two-phased process. The first phase is characterized by a rapid and high peak, while the second phase is much slower and has lower levels of insulin secretion. It is proposed that such kinetics are associated with the availability of insulin granules in beta cells during the first stage and recruiting granules from internal storage during the second one [50].

In native islets, insulin secretion is synchronized, robust, and rapid [51]. Stem cell-derived beta cells are different in terms of the time needed for response, graduality, and magnitude [32,33,34]. Because of these differences, the latter can be called functionally immature. Therefore, it is important to understand what changes after birth lead to final maturation of beta cells and the appropriate GSIS dynamics. During the first three weeks post birth, the expression of mature beta cell markers such as MAFA [52] and Urocortin3 (UCN3) [53] intensifies. However, changes in insulin secretion happen during the first week after birth (i.e., prior to modifications in gene expression) and cannot be accountable for that. Moreover, depletion of Ucn3 in mice does not affect functional maturation of beta cells [53]. What is important to notice is that the type of nutrition changes after birth: from placental in utero to intermittent postnatal feeding. Beta cells play a key role in the metabolism and adapt to this nutritional shift. Postnatal insulin secretion is stimulated by glucose, while fetal insulin secretion depends on amino acids [54]. The nutrient sensitivity of the mTORC1 (mammalian target of rapamycin complex 1) pathway plays a significant role in beta cell functional maturation. In stem cell-derived insulin-producing cells, higher sensitivity of mTORC1 enhances GSIS [54]. Another marker of adult but not neonatal beta cells is estrogen-related receptor γ (ERRγ). Through a transcriptional network, ERRγ affects ATP levels in cells that are required for GSIS. Overexpression of ERRγ promotes stem cell-derived beta cell maturation by intensifying insulin secretion [55]. Many regulatory pathways are necessary for the establishment of GSIS. Inhibition of transforming growth factor beta (TGF-b) signaling is required during differentiation of endocrine progenitor cells, but further TGF-b signaling is needed to develop mature functional beta cells. The more mature cells display biphasic insulin secretion [56].

DNA methylation and histone modification are key factors for pancreatic cell differentiation and functions [57]. Pancreatic alpha and beta cells have definitive open chromatin regions which are connected to cell-specific gene expression [58]. Epigenetic modifications in enhancers are crucial for cellular commitment during development. In pancreatic cell differentiation, poised enhancers interact with pioneer factors FOXA1 and FOXA2 (forkhead box A1/2). This complex is recognized by the linage-specific transcriptional factor PDX1, which activates pancreatic genes. These interactions between enhancer regions, pioneer factors, and linage-specific transcriptional factors result in the acquisition of competence and pancreatic cells specification [59]. It is known that there is a connection between metabolism and epigenetics—together they regulate endoderm differentiation from PSCs [60]. For instance, Crotonyl-CoA, which plays an important role in fatty acid and amino acid metabolism, is involved in histone acetylation [61] and promotes entoderm specification [60]. How metabolic changes affect alterations in specific chromatin regions not only during entoderm differentiation but also in islet cells is a potential objective of further research.

The cytoskeleton can also affect the differentiation and function of beta cells generated from stem cells. The depolymerized state of the cytoskeleton promotes differentiation of NGN3^+^ cells and leads to improved GSIS dynamics [62]. Adding compounds that modulate the cytoskeleton to mimic conditions of a developing pancreas in vivo can be another approach to obtain mature insulin-producing cells and enhance GSIS. Considering another component of the cytoskeleton, microtubules (MTs) in murine islets regulate insulin secretion by altering their states [63]. Highly stable MTs suppress GSIS, while depolymerization and hyper-stabilization of MTs promote secretion from individual beta cells. Trogden et al. [63] also uncovered that depolymerization of MTs creates “hot spots” or “secretion clusters” of beta cells, which advocate for the timely response of GSIS, and proved the functional heterogeneity of insulin-producing cells. MTs regulate which secreting granules (the older ones or newly synthesized) are released, which may have an unidentified effect on GSIS [64].

Better understanding of epigenetic, molecular, and cellular characteristics that are necessary for normal insulin secretion is crucial for producing functional beta cells in vitro. To further address this issue, it is important to investigate different signaling pathways and regulators of GSIS whose activity may change with the islet maturation and after birth.

## 4. Further Steps

### 4.1. Organoids

The intensity and response time of GSIS in stem cell-derived beta cells may also differ from those in native beta cells because of the lack of cellular interaction during differentiation and maturation. As beta cells develop and function in vivo in islets consisting of various cell types, it is logical to speculate that recreating similar conditions in vitro will improve the differentiation and maturation of beta-like cells.

Creating islet organoids that would comprise all endocrine cell types can be the next step. These cell types can be differentiated similarly to beta cells from hPSCs or hESCs. For instance, alpha cells affect GSIS and play a role in diabetes [65]. It might be promising to couple stem cell-derived alpha cells [66] with beta cells during or after maturation. It has been shown in mice that immortalized alpha- (α-TC6-1) and beta cell (MIN-6) lines in a co-culture interact with each other. MIN-6 inhibited glucagon gene expression through a molecular mechanism that is still to be identified [67]. Another study based on murine alpha and beta cells showed the role of the glucagon receptor on beta cells in GSIS [68]. Performing similar research using human stem cell-derived alpha and beta cells can improve the understanding of cellular and molecular interactions between them and might lead to their potential use in producing insulin-secreting cell organoids with more synchronized and robust insulin release.

Hormones secreted by endocrine islet cell types act as paracrine regulators of beta cells. Their effects on insulin secretion can be negative. Somatostatin secreted from pancreatic delta cells is an inhibitor of alpha and beta cells. It is an agonist of G protein-coupled receptors (GPCRs) and acts as an inhibitor of insulin secretion [69]. It has been reported that somatostatin synthesis is intensified in diabetes [70]. Pancreatic polypeptide produced by PP cells inhibits somatostatin expression and glucagon release [71,72]. Exogenous ghrelin that is normally secreted by a small number of pancreatic epsilon cells acts as a suppressor of the cAMP signaling pathway, which reduces GSIS through several interactions [73]. On the other hand, the use of antagonists of growth hormone (GH) secretagogue receptor (also known as ghrelin receptor (GHS-R)) increases insulin release [74].

Co-cultures of human delta, PP, epsilon, and beta cells have not been analyzed. Because hormones released from the other endocrine islet cells can act as negative regulators of insulin secretion and are involved in disease, it is difficult to predict what the outcome of combining them together with stem cell-derived beta cells might be. In healthy native islets, all these cells are finely tuned together to maintain homeostasis and communicate with each other at multiple levels with positive and negative feedback. Recreating all these interactions is a complicated task and would require more research into cellular crosstalk and regulation.

To mimic the conditions under which beta cells exist, it is important to remember that beta cells exhibit morphological, functional, and molecular heterogeneity. Dorrell et al. [75] proposed that beta cells can be subdivided into four subpopulations based on cell surface markers such as ST8SIA1 (ST8 alpha-N-acetyl-neuraminide alpha-2,8-sialyltransferase) and CD9. Moreover, a decrease in beta cell heterogeneity can be associated with T2D [75]. Understanding how this affects islet viability and insulin secretion is important for a better insight into cell crosstalk and identification of the changes in diabetic patients [76].

Beta cells are highly affected by their microenvironment. It modulates gene expression, proliferation, metabolism, survival, and death. The beta cell microenvironment includes other endocrine cells, endothelial cells, extracellular matrix, pericytes, nervous cells, and immune cells [77]. Islets are highly vascularized, and that is why beta cells come in close contact with the endothelial cells and pancreatic pericytes. In native islets, beta cells are polarized. They connect to basement membrane proteins through endothelial cell laminins and beta cell b1-integrins. This crosstalk between endothelial cells and beta cells plays an important role in insulin gene expression and proliferation [78]. Beta cells form focal adhesion where they contact the extracellular matrix, which promotes insulin granule fusion [79]. hPSC-derived beta cells display enhanced insulin secretion when cultured on the basement membrane protein matrix [80]. Pancreatic pericytes regulate islet blood flow, and the crosstalk between these cells and beta cells appears to be important for glucose metabolism. It is also impaired in T2D [81]. Tissue-resident macrophages are known to play an important role in normal islets and in disease [82]. They can be important for beta cell proliferation, homeostasis, and function [83]. Inflammatory macrophages are involved in diabetes pathogenesis [84].

Pancreatic islets are innervated by sympathetic and parasympathetic nerves of the autonomic nervous system [85]. Sympathetic nerves produce neurotransmitters, such as noradrenaline, which inhibit GSIS and intensify glucagon release [86,87]. Parasympathetic neuropeptides such as vasoactive intestinal polypeptide (VIP) [88] and pituitary adenylate cyclase activating polypeptide (PACAP) [89] stimulate insulin and glucagon secretion. It has been shown that in vitro treatment of human islets with PACAP and their subsequent transplantation in mice improved cell viability and intensified insulin secretion. However, blood glucose levels were not affected, probably due to joint intensification of glucagon release from alpha cells [90]. Nevertheless, pre-treatment of a graft with PACAP could be implemented in research for the best islet transplantation method. Insights into how the nervous system affects beta cells and islets in general can be useful for studying the regulation of insulin secretion and in vitro construction of islet organoids. All the components of the microenvironment, either separately or together, affect beta cells and form a functional unit (Figure 2). More knowledge about this crosstalk will allow scientists to understand how beta cells are involved in it and what dysfunctions lead to diabetes and other disease.

### 4.2. Transplantation

When researchers finally produce fully functional beta cells or islet-like structures, another question is how to safely deliver and maintain grafts in recipient patients. To effectively restore normoglycemia, stem cell-derived insulin producing cells need to be transplanted and retain their viability and function for a long-term period.

Islet cell loss after transplantation is associated with poor vascularization [91] and inflammation [92]. To achieve a more successful outcome, islet cells could be combined with other cell types to reduce cell death. One study showed that a combination of human islet cells with human amniotic epithelial cells (hAECs) improved graft viability and maintained cell functions after transplantation into diabetic mice [93]. hAECs secrete proangiogenic and anti-inflammatory growth factors, which protect islet cells from cell death and promote angiogenesis in transplanted grafts. It is known that mesenchymal stem cells (MSCs) produce such regulatory molecules as interleukin-6 (IL-6), vascular endothelial growth factor-A (VEGF-A), hepatocyte growth factor (HGF), and TGF-b that increase cell survival and promote vascularization [94]. Co-culturing of pancreatic islets with MSCs has shown positive results in terms of cell viability [95]. Rat MSCs co-cultured with pancreatic islets expressed endocrine progenitor marker Pdx1 and over time began to release insulin [96]. Interestingly, in a clinical trial, early-stage diabetes patients that were treated with MSCs showed slower diabetes development [97]. These properties of MSCs could be applied in generating stem-cell derived insulin-producing cells and their subsequent transplantation.

The combination of hiPSC-derived beta cells with human adipose-derived stem cells (hADSCs) and human umbilical vein endothelial cells (HUVECs) in 3D organoids promotes insulin secretion most probably by intensifying the ERRγ pathway. hADSCs could act as cells similar to pancreatic fibroblasts, and HUVECs could represent endothelial cells that promote vascularization. The WNT pathway in hADSCs is enhanced, which also positively affects beta cells [98]. It is known that expression of WNT4 intensifies in maturating islets after birth [99]. Such cells as hAECs, hADSC, HUVECs, and MSCs could be potentially transplanted with stem cell-derived beta cells, as their regulatory signals promote maturation and intensify GSIS. How these cells will affect other endocrine cell types in islet organoids is unknown, and this fact provides further research possibilities.

Much attention has been drawn to extracellular vesicles (EVs) and their role in the cellular crosstalk and disease [100]. EVs, and exosomes as their subset, contain long non-coding RNAs (lncRNAs), microRNAs (miRs), mRNAs, DNA fragments, proteins, metabolites, and more [101]. Beta cells produce exosomes that play a role in preserving pancreatic islet architecture and its function and in inducing islet angiogenesis, which also implies that treatment with exosomes could be used as a novel therapeutic strategy for diabetes [102] or for achieving better transplantation outcome. Murine islet-derived exosomes include lncRNA-p3134, which in MIN6 cells positively regulates GSIS by intensifying the expression of Pdx1, MafA, glucose transporter 2 (Glut2), and transcription factor 7-like 2 (Tcf7l2), as well as the PI3K/Akt/mTOR pathways [103]. MicroRNA-26a (miR-26a) influences beta cell proliferation and insulin secretion in an autocrine manner and can affect peripheral insulin sensitivity in a paracrine manner [104]. Not all the exosomes’ components act as positive regulators in beta cells. The miRNA-29-TNF-receptor-associated factor 3-mediated pathway can be the cause of macrophage recruitment to islets and inflammation, so exosomal miR-29 that can be produced in beta cells is associated with T2D [105]. Research into beta cell-derived exosomes and their contents could provide more information about the cell-to-cell islet communication. Beta cell proliferation, insulin secretion, and angiogenesis in grafts islet organoids could be regulated by adjusting positive regulatory signals or blocking the negative ones. In addition, screening exosome profiles in diabetes can be a tool for early detection of the disease.

Transplanted beta cells derived from non-patient-specific stem cells or islet organoids would require immunosuppression or encapsulation in a non-immunoreactive capsule. The latter is a more prospective approach, as the long-term use of immunosuppressants might display major side effects [106]. Choosing an appropriate material is a challenge. Alginate-derivatives can be a promising option to encapsulate insulin-producing cells [107]. The function of such capsules is not only to protect cells from immune system response, but also to allow nutrients and oxygen to be transferred inside as well as insulin and other by-products to be released [108]. It is possible to develop hydrogel models that could encapsulate islets and supporting cells such as endothelial cells or MCSs, as well as ECM and growth factors [109]. Hydrogels are permeable for blood vessels, so cells will be able to access blood oxygen. A major drawback is the high probability of a blood-mediated inflammatory reaction, which will ultimately result in graft cell death [109]. There are some ways to avoid that: for example, by incorporating anti-inflammatory peptides in hydrogel capsules [110]. Materials should also be mechanically resistant and preserve their integrity over a long period of time [108]. All this shows that islet encapsulation methods must be improved for best transplantation outcomes.

Beta cell elimination in patients with T1D is caused by autoimmune attack mainly mediated by T cells. This could be a major complication in stem cell-derived beta cell therapy. It is difficult to assess the effect of diabetogenic T cells on beta cells derived from hPSCs or hESCs because animal models of T1D do not fully resemble this condition in humans. Usage of encapsulation techniques can mitigate the risks of CD8 T cell cytotoxic attack, but transplantation of “uncovered” beta cells could provide better vascularization and therefore a higher graft survival rate. Programmed death-ligand 1 (PD-L1) is known to be upregulated in islet cell transplants [111]. hPSC-derived beta cells that lack HLA (human leukocyte antigen) class I molecules and overexpress PD-L1 are less affected by diabetogenic T cells than unmodified cells [112]. There is a possibility to generate stem cell-derived beta cells that would be less susceptible to autoimmune response.

Finding an appropriate method to protect the transplant from immune system, identification of the accessible location and predictions of the risks, as well as the ways they can be avoided are some objectives that would need to be addressed before clinical trials involving stem-cell derived beta cells.

## 5. Conclusions

Diabetes is a pressing global issue. In this review, we imply that stem cell therapy, which involves using PSC- and ESC-derived insulin-producing cells, would be a better option than methods that are available now. There seem to be a few complications in the development of an effective protocol for generating mature beta cells. There is probably no simple answer or way to influence the differentiation process to finally produce stem cell-derived beta cells identical to the native ones. There must be a complex understanding and implementation method. To address these issues, several promising areas of research were identified. One field is the metabolic maturation of beta cells and metabolic–epigenetic regulation. Current protocols of endoderm and pancreatic differentiation from hESCs and hPSCs mostly focus on the influence of epigenetic regulators alone and overlook the importance of metabolism. Acquisition of a metabolic program for pancreatic beta cells could be connected with a specific chromatin state, especially in regulatory regions, and histone modifications. It was mentioned that in beta cells, insulin secretion stimuli change in the postnatal period [54]. Therefore, functional maturation of beta cells might be related to the changes in epigenetics and metabolism. It could be important to identify epigenetic landscapes during each stage of beta cell differentiation in vivo and then to compare it with in vitro differentiation of insulin-producing cells from ESCs and PSCs, as well as research epigenetic–metabolic links. It is possible that the components of important regulatory pathways are not activated in the process of stem-cell derived beta cell maturation in vivo due to the lack of cell predisposition in one or several stages or due to the absence of environmental cues. The same reason might account for the presence of polyhormonal cells—commitment to beta cell identity is not fully achieved. Hence, it might be relevant to compare epigenetic competence of cells in each stage of differentiation.

Pancreatic islets are complex multicellular structures comprised of various cell types including non-endocrine ones, which in many ways affect and regulate cell polarity, gene expression, and all cellular functions. That is why it is incorrect to neglect these other components and focus only on beta cells. Perhaps creating islet organoids with other stem cell-derived endocrine cells, supporting cells, and ECM components could promote the maturation of beta cells. Members of the cellular crosstalk such as EVs account for beta cell function, and their implementation in differentiation is another branch of research. Delivery of beta cells or islet organoids is also an objective of further investigations in regenerative medicine and diabetes treatment. With all the available information on beta cells in health and disease, we still need to make some more effort to develop stem cell-derived beta cells identical to the native ones.

## Figures and Tables

**Figure 1 biomolecules-12-00810-f001:**
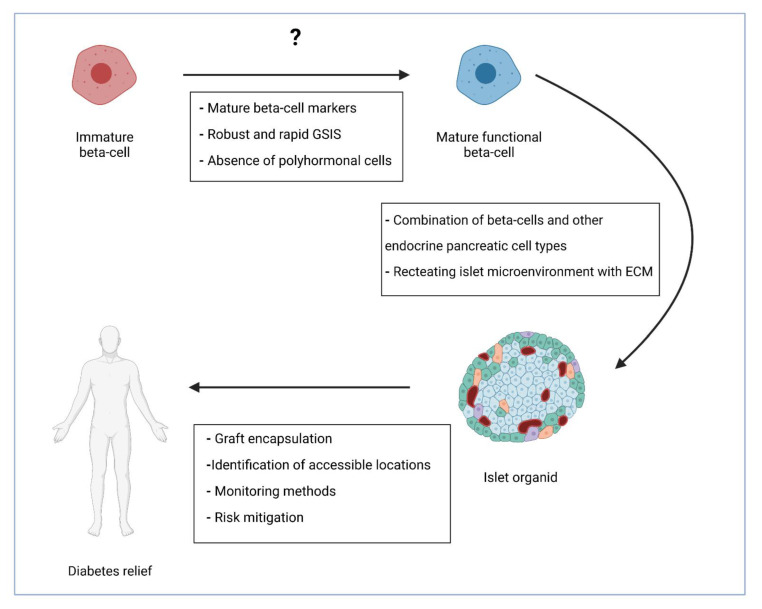
Further steps in cell-based diabetes treatment. Generation of fully mature insulin-producing cells that are functionally identical to native beta cells is the first stage. The next one could be the combination of beta cells with supporting cells and encapsulation of such islet organoids. The final stage would be finding the best transplantation method with minimal risks. GSIS—glucose-stimulated insulin secretion; ECM—extracellular matrix. Created with BioRender.com.

**Figure 2 biomolecules-12-00810-f002:**
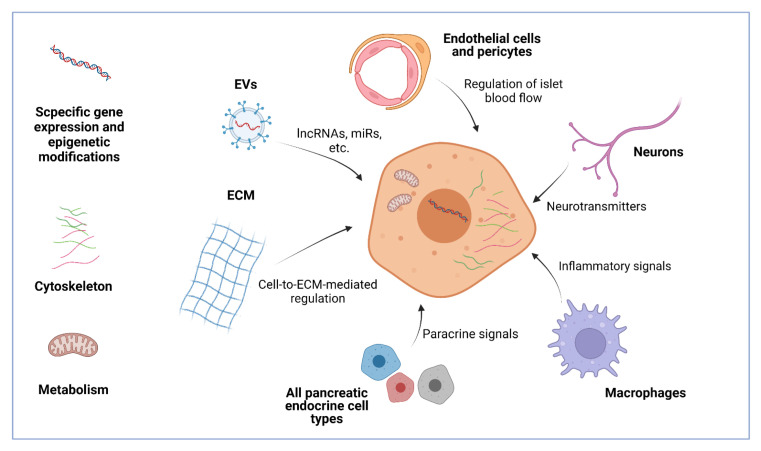
The microenvironment of beta cells includes many components such as pancreatic islet endocrine cells, endothelial cells, pericytes, neurons, macrophages, the extracellular matrix (ECM), and EVs (extracellular vesicles). All of them send different signals to beta cells, affecting their viability, proliferation, gene expression, and functions such as GSIS. These effects can be both negative and positive. Intracellular regulatory signals include genetic and epigenetic cues, cytoskeleton, and metabolic states. By intensifying some regulators and blocking others, it is possible to promote maturation, GSIS, and graft survival, which are all crucial for stem cell therapy of diabetes. LncRNAs-long non-coding RNAs; miRs-microRNAs. Created with BioRender.com.

**Table 1 biomolecules-12-00810-t001:** Methods to generate beta cells from various cell sources. Direct differentiation protocols.

Method to Generate Beta-Cells	Advantages	Disadvantages	References
Progenitor cells	Pancreatic stem-cell population, self-regeneration of pancreatic islets	No direct evidence of presence in humans	[37,38,39]
Transdifferentiation	Cell availability, possibility for personalized cell therapy	Low differentiation efficiency, high cell heterogeneity, functional immaturity, low GSIS, absence of key beta cell markers	[17,18,19,20,21,22,23,24,25,26]
Direct differentiation (ESCs, PSCs)	Usage of undifferentiated cells, mimicking embryological development, relatively robust GSIS, low heterogeneity, possibility for personalized cell therapy	Signs of functional immaturity, differences in metabolic pathways	[32,33,34,35,36]
**Direct differentiation protocol**	**Advancement**		
D’Amour et al., 2005 [16]	Efficient differentiation of hESCs to definitive endoderm		
Chen et al., 2009 [29]	Generation of pancreatic PDX1^+^-progenitors from hESCs		
Nostro et al., 2011 [30]	Roles of TGF-b, WNT, nodal/activin A, and BMP (bone morphogenetic protein) signaling in the pancreatic lineage cell specification		
Rezania et al., 2014 [32]; Pagliuca et al., 2014 [33]	Generation of stem cell-derived beta cells closely resembling native beta cells		
Russ et al., 2015 [34]	Omission of BMP inhibitors during pancreatic specification results in higher yield of PDX1^+^/NKX6.1^+^-cells		
Nair et al., 2019 [35]	Endocrine cell clustering mimics pancreatic organogenesis and promotes beta cell differentiation		
Liu et al., 2021 [36]	New combination of chemicals and 3D pancreatic progenitor clusters promote beta cell functions		

## Data Availability

Not applicable.

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
