# Peer review of "Overcoming the Limitations of Stem Cell-Derived Beta Cells"

_biomolecules, 2022, doi:10.3390/biom12060810_

Round 1
Reviewer 1 Report
The review article by Karimova et al focuses on the current progress in generating functional human pancreatic islets and the limitations hindering the use of hPSC-derived β cells for diabetes treatment. Overall, the review is well written and topic is very important. There are some points that should be addressed as follows:
Major comments,
1) It is better to add a table summarizing the current limitation for stem cell derived beta cells. The table should include the methods (direct differentiation, trans differentiation, progenitor expansion etc.
2) Stem cell derived beta cells can undergo functional maturation in vivo within a few weeks to a few months. Therefore the statement “Another reason why cells do not fully maturate might not be related to the last stages. It may be that stem cells at the very beginning of differentiation do not gain full endoderm commitment, so at later stages they do not respond to the regulatory signals. “ may not be represented current understanding of functional maturation of stem cell derived beta cells. The authors should highlight the truth Stem cell derived beta cells can undergo functional maturation in vivo.
3) Similar review papers relate to functional maturation of stem cell derived beta cells were published this year (PMID: 35244743; PMID: 35557947; PMID: 34995481). Author should reference these review to emphasize the importance of overcoming the limitations of stem cell derived beta cells.
Author Response
We would like to thank you for these valuable critiques and recommendations. We took all your comments into account and made corrections according to them. Please see the attachment.

Reviewer 2 Report
This review by Mariana V. Karimova et al. describes the importance of regeneration of stem cells derived beta cells that could act as an unlimited source of insulin-secreting cells ultimately benefiting type 1 type 2 diabetes patients. In this review, they mainly focus on the development and current state of various protocols, consider advantages, and provide insights into how their disadvantages are been overcome. Also, they emphasized the different and new methodology to generate insulin-producing beta cells. The topics covered in this review are quite convincing. While this information, in general, is of interest and add to the current literature illustrating the role of stem cells in producing beta cells and another essential cell type against several diseases, there are several reviews already have been published discussing the importance of stem cells. I have some concerns and suggestions regarding the literature and presentation.
Major critiques:
· The abstract need to be rewritten, some of the lines are not very clear or look incomplete (e.g page 1, lines 12-13).
· Precisely, the generation of the stem cells derived beta cells are useful, while these cells can also be targeted by the immune system which has not been discussed in this review. It is important and advisable to enlighten the strategies that protect stem cell-derived pancreatic beta-cells from the immune system or diabetogenic t cell recognition which are already in circulation in T1D patients and their cytotoxic attack.
· Is it possible to measure T cell responses towards stem cell derived beta cells in strictly human autologous settings?
· What is the fate of human HLAs and their expression in these stem cells-derived beta cells?
· Does the author find and worth discussing the difference in the secretion of insulin between stem cells-derived beta cells and primary adult β cells etc.?
· Does the author think distinguishing the type of stem cells-derived beta cells in their function as two different types of beta cells in human diabetes as an adult vs juvenile human islet?
· Are their studies or the robust disease animal models discussing the successful implementation of stem cell-derived beta cells? Should be included in this study.
· What do authors think of the relation between the stem cells-derived beta cells and stem-like T cells in type 1 diabetes?
· Minor critiques:
· T1D and T2D are used in the abstract with no explanation, abbreviations should be avoided in the abstract.
· It is advisable to provide the outline of stem cell-derived islet differentiation protocols in tabular format which would be easy to grasp for the readers.
· It is advised that the authors should define abbreviations at least in the first instance PDX1, NGN3, MAFA, PAX6, PAX4 (page 2, line 58).
· There are a lot of definite and indefinite articles missing along with some grammatical errors.
· Also please correct the manuscript for undefined spaces, an editor is required to correct these.
Author Response

(The authors gave the same response as above.)
